# Assessing the impact of the global subsea telecommunications network on sedimentary organic carbon stocks

M. A. Clare[1] ✉, A. Lichtschlag[1], S. Paradis [2] & N. L. M. Barlow [3]

The sequestration of organic carbon in seafloor sediments plays a key role in regulating global climate; however, human activities can disturb previously-sequestered carbon stocks, potentially reducing the capacity of the ocean to store $CO_2$. Recent studies revealed profound seafloor impacts and sedimentary carbon loss due to fishing and shipping, yet most other human activities in the ocean have been overlooked. Here, we present an assessment of organic carbon disturbance related to the globally-extensive subsea telecommunications cable network. Up to 2.82–11.26 Mt of organic carbon worldwide has been disturbed as a result of cable burial, in water depths of up to 2000 m. While orders of magnitude lower than that disturbed by bottom fishing, it is a non-trivial amount that is absent from global budgets. Future offshore developments that disturb the seafloor should consider the safeguarding of carbon stocks, across the full spectrum of Blue Economy industries.

Marine sediments are the largest store of organic carbon on Earth and this sequestration plays a key role in regulating global climate[1–4]. However, if previously-buried organic carbon stocks are disturbed and exhumed, this can lead to remineralization of carbon to $CO_2$ (which could potentially increase ocean acidification), limiting the capacity of the ocean to store additional $CO_2$, and potentially adding to the build-up of atmospheric $CO_2$[3–6]. Sedimentary carbon stocks can be episodically disturbed by natural events, such as floods, storms that resuspend shallow seafloor sediments, or large earthquake-triggered submarine landslides[7–11]. In addition to these natural events, human activities that impact the ocean floor (e.g. fishing, mining, oil and gas exploration, aggregate extraction, anchoring) are increasingly recognized as playing a significant role in the release of previously-buried organic carbon, with intensity and spatial extent growing by the increased use of marine resources and Blue Growth[2–4,12–16]. It is estimated that 1.3% of the global ocean-floor is trawled each year (~$5 \times 10^6$ km$^2$), potentially releasing similar quantities of sedimentary organic carbon to agricultural tillage on land[17]. To what extent other human activities release previously-buried carbon remains unclear; largely due to lack of access to industry datasets that permit quantification of that disturbance. This limitation inhibits assessment of the

impacts of the full extent of human activities on carbon burial efficiency worldwide. Here, we assess the potential impact of one of the most extensive infrastructure systems on our planet—the network of subsea telecommunications cables that span more than 1.8 million km across the global ocean (Fig. 1).

More than 99% of all international digital data traffic is routed via >400 interconnected submarine cable systems (Fig. 1A), which underpin the Internet, enable remote working, financial transactions worth trillions of dollars per day, and connect remote island states to sustain their economic development[18,19]. These cables, which are either laid directly on the seafloor or buried and typically have a diameter equivalent to a garden hosepipe (but may increase to 4–5 cm diameter in shallow water to accommodate integral steel wire armouring for protection), are vulnerable to damage by external threats that can halt connections and/or significantly reduce bandwidth, requiring expensive and logistically-challenging repairs. Analysis of a global industry database revealed that approximately 150–200 cable faults occur each year, with most (60–70%) caused by human activity in <200 m water depth[18]. The main causes are fishing (41% of faults) and accidental anchor drops from vessels (16%). Bottom trawling is the most common type of fishing to interact with submarine cables as it occurs on most

[1]Ocean Biogeoscience Research Group, National Oceanography Centre, Southampton, UK. [2]Geological Institute, ETH Zürich, Zürich, Switzerland. [3]School of Earth and Environment, University of Leeds, Leeds, UK. ✉e-mail: michael.clare@noc.ac.uk

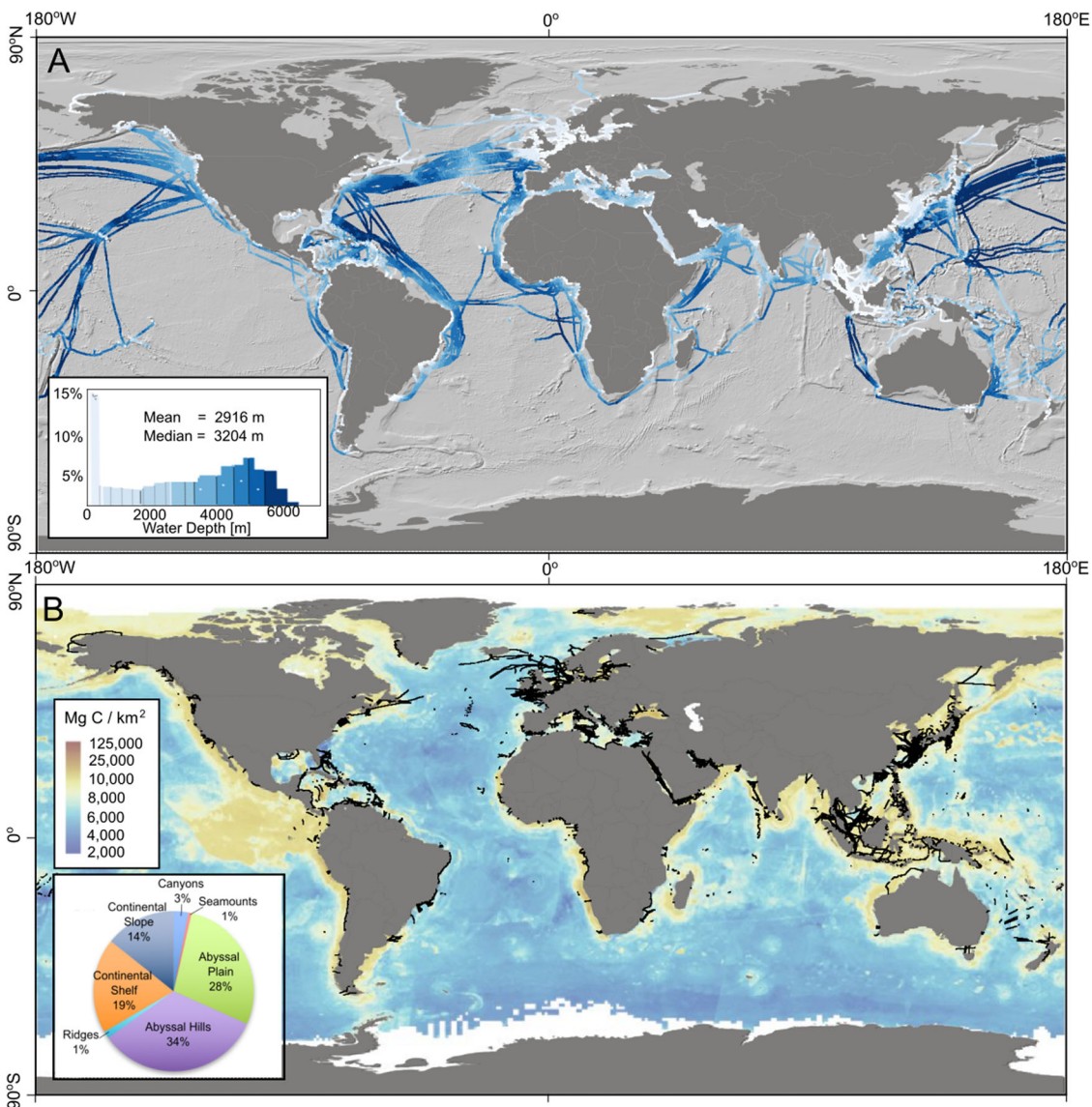

**Fig. 1 | Extent of submarine telecommunications cables across the ocean and seafloor sedimentary carbon stocks. A** Cable routes are colour coded (white to dark blue) according to water depth (m). Bathymetry derived from the GEBCO_2022 Grid, GEBCO Compilation Group (2022) GEBCO 2022 Grid. **B** Extent of cable routes in water depths of up to 2000 m illustrated as black lines, overlying global distribution map of sedimentary carbon stocks in the first meter below the seafloor from Atwood et al.[2]. Inset pie chart shows the relative length of all cable routes that cross different physiographic domains, based on global geomorphic mapping[63]. Country outlines sourced from Natural Earth free vector data.

continental shelves and covers large areas of seafloor[13,18–20]. In areas of such potentially-damaging human activity, cables are buried for protection by intrusive trenching, ploughing or jetting techniques[21] (Fig. 2). Cable faults caused by bottom trawling typically relate to the dragging of heavy (0.1–8 tonne) otterboards that can penetrate tens of centimeters into soft sediments and snagging of weights that are designed to stir up the seafloor to capture fish and shellfish[18,22]. The depletion of fishing stocks (largely driven by overfishing) has stimulated a push of demersal fishing into deeper waters in several regions[15,23], meaning that cable burial is increasingly required on parts of the continental slope (in water depths up to 1500 m), in addition to the continental shelf. In some areas (e.g. north-east Atlantic and eastern Pacific Ocean), cables in up to 2000 m water depth may be buried, due to the expansion of fishing activity to greater water depths[24]. In deep water, where fishing and other disturbance activities are rare (i.e. less than four cable faults occur in the High Seas per year[19]), telecommunications cables are unarmoured and laid directly on the seafloor, causing only very minor disturbance of sediments[25–27].

Previous studies have investigated the environmental interactions of submarine telecommunications cables, concluding that they typically exert benign to minor physical impacts on seafloor ecology[25–27]. However, it has been shown recently that disturbance by offshore human activities, such as trawling, aggregate dredging and anchoring can remove previously-buried carbon from seafloor sediments[16,17,28]. To date, however, no study has considered the volumes of sediment and contained organic carbon disturbed as a result of cable burial, especially considering that greater depths (i.e. up to 2 m) below seafloor than fishing activity will be disturbed. Here, we aim to assess this issue at a global scale with a view to informing more efficient management strategies to minimize future carbon release. We do this by addressing the following questions. First, what is the global footprint of seafloor disturbance by cable burial, and what is the total volume of sediment that has been disturbed by cable installation to date? Second, what volume of organic carbon has been disturbed by cable burial, and what is the likely loss of previously sequestered carbon as a result of cable burial activity? Third, how do the disturbed volumes of

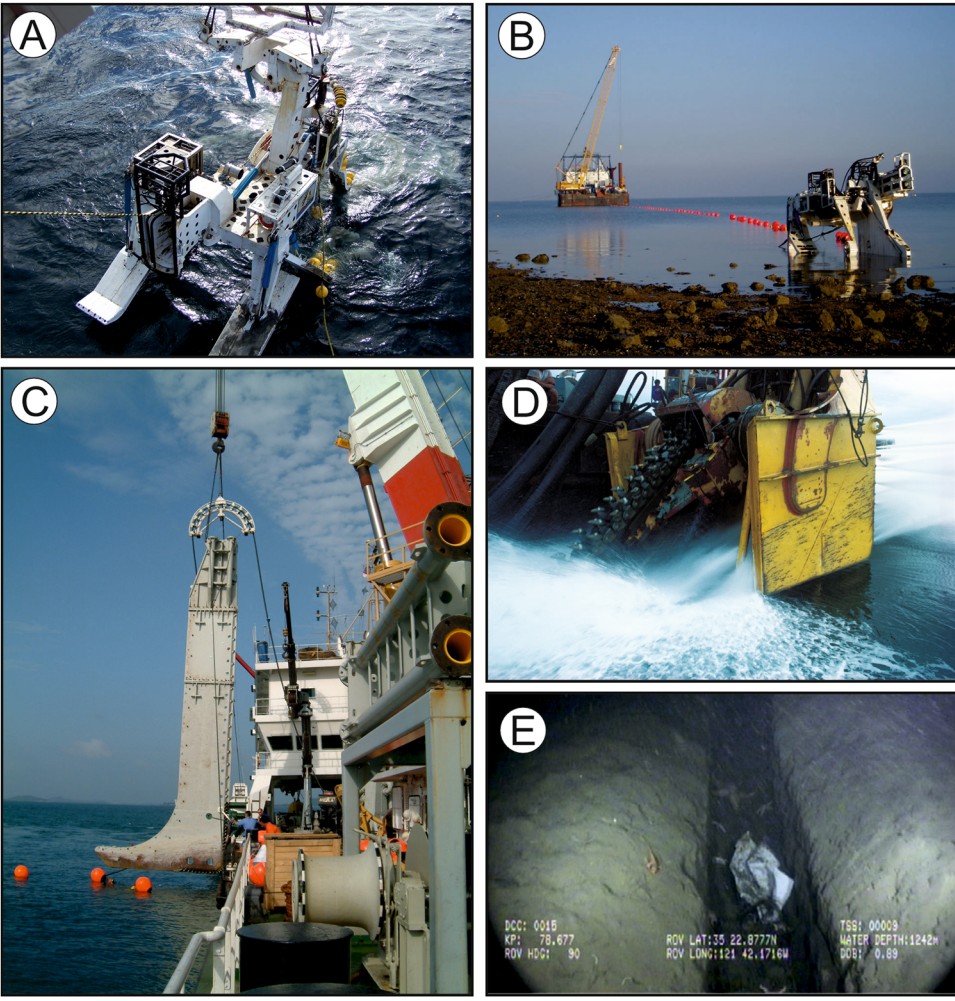

**Fig. 2 | Various devices used to excavate trenches for cable burial.** Devices include **A**, **B** plough, **C** jet, and **D** mechanical trencher. A photograph of a steep-sided 0.5 m-wide trench at 1242 m water depth is shown in **E** that was excavated using a jet in consolidated cohesive sediments (modified from[21]). Photographs **A**–**D** courtesy of Global Marine Group.

sediment and organic carbon compare to other natural processes and human activities? Finally, where is organic carbon more vulnerable to cable burial based on regional hotspots of reactive organic carbon?

In this present study we report a global assessment of the impacts of cable burial on sedimentary organic carbon stocks through integration of a global database that documents the extent and locations of submarine telecommunications cables, with a modelled distribution of organic carbon hosted in modern ocean sediments worldwide (trained with >11,000 sampling points)[2,29]. We show how up to 2.82–11.26 Mt of organic carbon worldwide has been disturbed by cable burial, and place this figure in a wider context through comparison with natural processes and other human activities.

## Results
### Sediment volumes disturbed by cable burial activities
As a base case, we assume that cables on the continental shelf (accounting for 16% of the total cable length worldwide) and on the continental slope to a water depth of 1500 m (13%) are all buried (Fig. 3). Below these water depths, some, but not all cables are buried; hence, for conservatism we also include cables lying between 1500 and 2000 m water depth (5%); the remaining cable length (i.e. 66%) is laid directly on the seafloor and is not buried. We assume a range of burial depths (0.5–2.0 m) and width of seafloor disturbance[21] (0.5–1.0 m) (see Methods). Integrating these excavated dimensions indicates that the cumulative wet volume of sediment that could have been disturbed by

cable burial activities to date may be as high as 0.13–1.05 km$^3$ in water depths up to 1500 m (Table 1). Assuming disturbance extends to 2000 m water depth, yields an additional disturbed sediment volume of 0.02–0.17 km$^3$, which makes a total of 0.15–1.22 km$^3$ disturbed sediment (an average of 0.004–0.04 km$^3$ per year since the start of records).

### Disturbance of sedimentary organic carbon stocks
Published global estimates of sedimentary organic carbon typically focus on the top 5–10 cm below seafloor[30]; however, cable burial affects greater depths[21]. To infer the potentially disturbed stock of sedimentary carbon by cable burial activities, we use a global model that accounts for stocks within the first meter below seafloor[2] (Fig. 1B). In the absence of any global dataset that extends below one meter, we necessarily assume a similar concentration of organic carbon exists to a depth of two meters (i.e. the maximum depth of cable burial assessed here). We accept this may result in an over-estimated disturbed carbon stock for that lower meter, and this data gap clearly underlines a need for greater constraint by future studies. In this model, the median of carbon stocks across continental slopes worldwide is 8632 Mg/km$^2$, which is similar to that encountered along cable routes between 200–1500 m (8690 Mg/km$^2$) and 1500–2000 m (9087 Mg/km$^2$) water depth. The median value for carbon stocks across continental shelf sediments worldwide is 18,666 Mg/km$^2$ [2], yet the median value encountered along cable routes on continental shelves is less than half

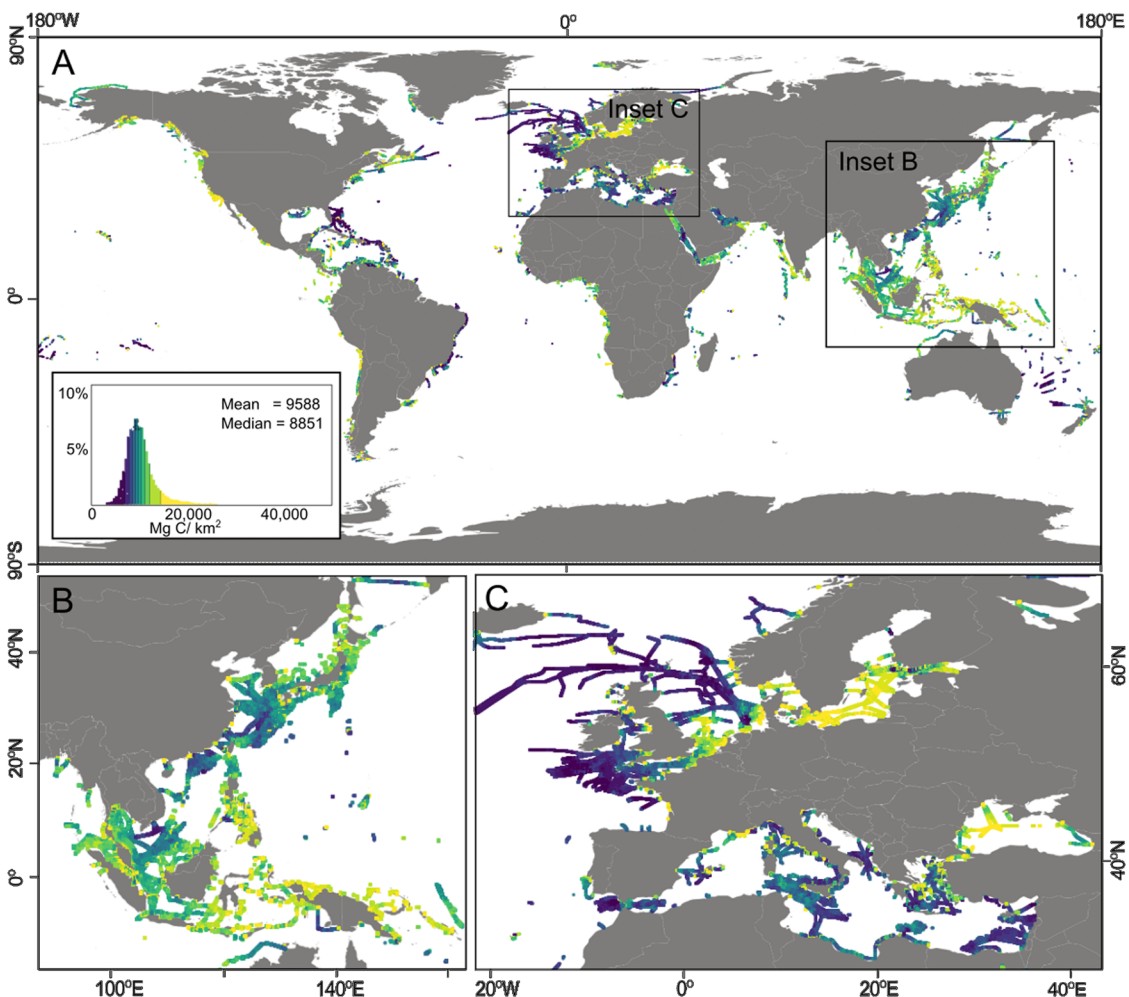

**Fig. 3 | Sedimentary carbon stocks in the first meter below the seafloor along lengths of cable routes where we assume cables are buried (i.e. 0–2000 m water depth). A** Overview global map and zoom in to regions that feature localized hotspots including (**B**) South-East Asia and (**C**) Southern North Sea, Baltic Sea and Mediterranean Sea. Country outlines sourced from Natural Earth free vector data.

**Table 1 | Length, area and volume of potentially disturbed sediment and organic carbon by cable burial activities**

| | Continental Shelf | Continental Slope to 1500 m WD | Continental Slope 1500-2000 m WD | Total |
|---|---|---|---|---|
| Length of cable [km] | 288,054 | 238,832 | 82,939 | 609,825 |
| Disturbed area [km²] | 144–288 | 119–239 | 41–83 | 304–609 |
| Disturbed sediment volume [km³] | 0.07–0.58 | 0.06–0.48 | 0.02–0.17 | 0.15–1.22 |
| Disturbed carbon stock [Mt] based on global average values from Atwood et al. (2020) for continental shelf or slope | 1.34–10.75 | 0.52–4.12 | 0.18–1.43 | 2.04–16.30 |
| Disturbed carbon stock [Mt] based on mapped values from Atwood et al. (2022) | 0.72–5.83 | 0.49–3.89 | 0.19–1.54 | 2.82–11.26 |

Minimum values are based on an excavated width of 0.5 m and a depth of 0.5 m, while maximum values are based on a width of 1 m and a depth of 2 m.
While cables are not consistently buried to 2000 m, this is the case in some regions; hence two water depth categories on the continental slope are considered. *WD* water depth.

that value, at 8880 Mg/km². Therefore, many of the global hotspots of sedimentary organic carbon on the continental shelf are not crossed by cable routes. We calculate disturbed carbon stocks based on mapped sedimentary carbon stocks[2] along cable routes. Assuming the most conservative burial scenario of up to 2000 m water depth, the estimated volume of disturbed sediment on the continental shelf and slope equates to a disturbed sedimentary organic carbon stock of between 2.82–11.26 Mt (Table 1), of which almost equal contributions come from disturbances on the continental shelf (51%) and the slope (49%). There is considerable geographic variability in carbon stocks

that may have been disturbed by cable burial activities (Fig. 3), particularly between different ocean basins (Fig. 4). The Baltic Sea stands out as the main region where cables intersect with the highest concentrations of sedimentary organic carbon, followed by the Pacific Ocean and South China and Eastern Archipelagic Seas that also feature high relative concentrations of carbon along cable routes (Fig. 3 and 4).

Determining how much of those disturbed stocks are lost (i.e. oxidized and not re-sequestered) is a far more challenging issue. Disturbance of the seafloor can physically remove organic carbon through erosion, which would subsequently be re-deposited elsewhere with

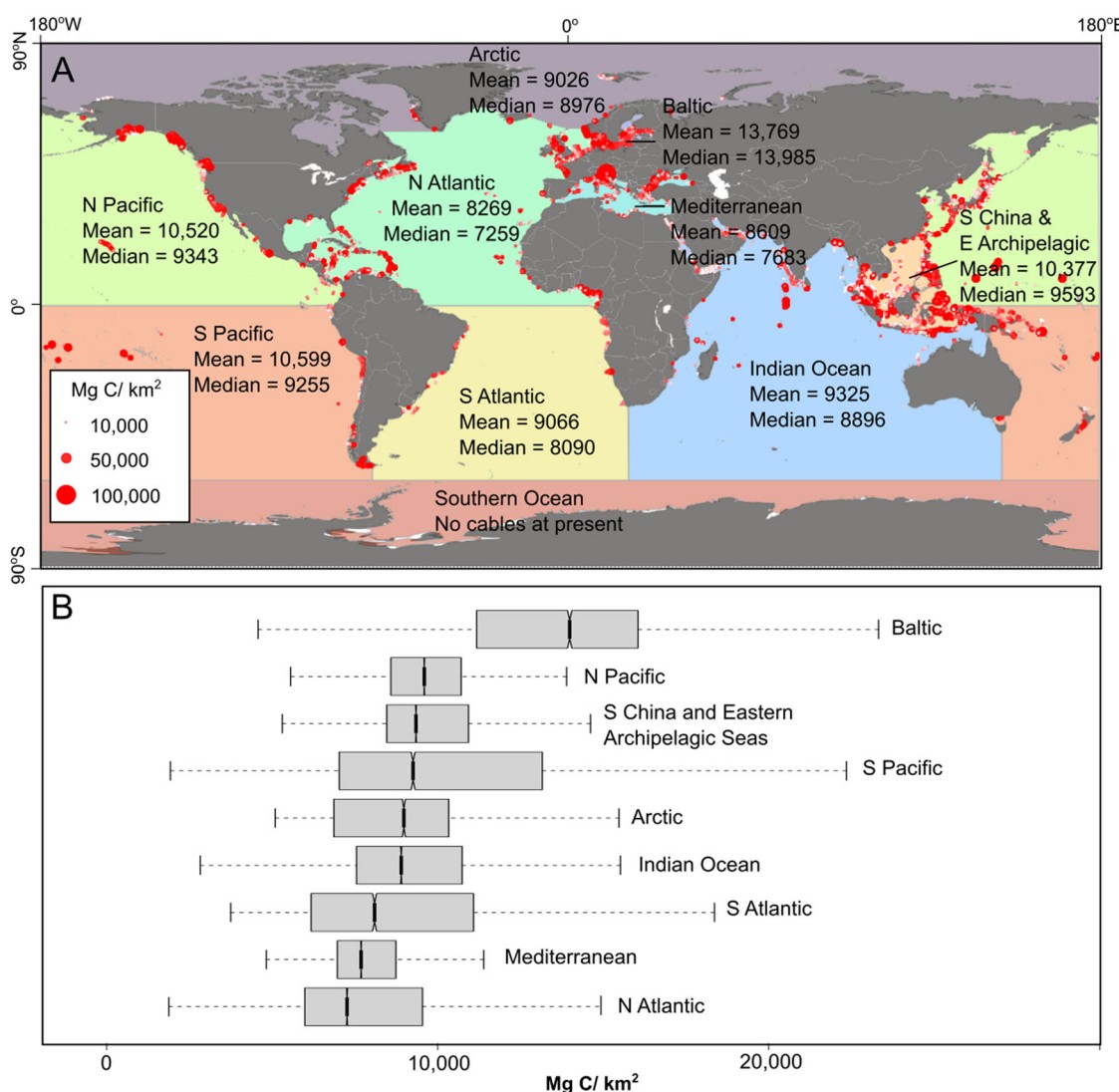

**Fig. 4 | Geographical distribution of carbon hotspots encountered by cable burial.** Sedimentary carbon stocks within top 1 m below seafloor along cable routes **A** shown where values are in the upper quartile of all values worldwide. Country outlines sourced from Natural Earth free vector data. Box and whisker plots **B** showing distribution of organic carbon stocks within top 1 m below seafloor along cable routes in different ocean basins of the world. Boxes show 25th and 75th percentiles with median annotate in between, while whiskers show full range of data.

little or no net loss of organic carbon, whereas another fraction of organic carbon can be oxidised and degraded into aqueous carbon dioxide as a result of the exposure of sediment to the oxygenized overlying water. Quantifying the organic carbon loss through each mechanism is complicated, and studies have attempted to estimate the release of organic carbon from bottom trawling disturbance, often with contrasting results[17,31–35]. Studies focused on the continental shelf and slope in the Mediterranean and North Sea found remineralization rates of 20–60% for seafloor organic carbon disturbed by deep sea trawling[17,31,32]. These remineralization rates were highest in areas affected by the greatest frequency of bottom trawls; however, as cable burial is a one-off activity, the highest remineralization rates are considered to be unlikely. Speculatively assuming the lowest loss rate (i.e. 20%) from these studies, would result in a cumulative loss of 0.144–1.17 Mt of previously-buried organic carbon on the continental shelf and 0.136–1.09 Mt on the continental slope (a total of 0.280–2.25 Mt globally). However, to date no study has specifically studied the effects of cable burial on carbon disturbance at field-scale and the whether findings from bottom trawling are truly applicable to cable burial remains unclear. Consequently, there remains considerable uncertainty in the fate of sedimentary organic carbon disturbed by cable

burial. First, not all excavated sediment will be released into the water column; instead most disturbed material likely rapidly resediments inside or close to the trench and thus may be effectively reburied, limiting potential for remineralization. Whether sediment rapidly backfills a trench will depend on the nature of the sediment (e.g. grain size), prevailing ocean currents near the bed and other background environmental conditions. In many cases it has been observed that trenches may refill within weeks to years on the continental shelf, but in some cases on the continental slope, where sediment supply is low this may take >15 years[21]. A particularly important control is likely to be the cable burial tool that is used, and the nature of the initial disturbance. In the case of ploughing and trenching, sediment typically settles quickly (particularly granular sediment, such as sand) and deposits close to the initial excavation site; in many cases immediately (fully or partially) backfilling the trench[21]. In such cases, the likelihood of remineralization will be reduced; however, in the case of jetting (which fluidizes the sediment), suspended plumes of fine (clay and silt-size) sediment may be more widely dispersed by ocean currents, taking days to settle and hence increasing the chances of remineralization[21,36]. Second, organic carbon mineralization rates will depend on external factors. For example, not all organic carbon stored

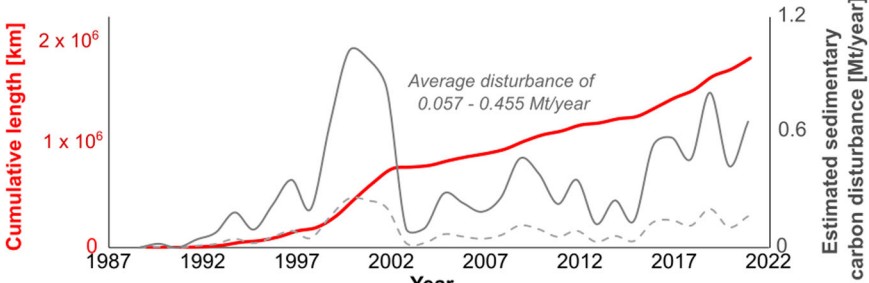

**Fig. 5 | Cumulative length of fiber-optic telecommunications cable installed (red) since start of records used in this study and the inferred rate of disturbance of sedimentary carbon (grey solid line is upper bound, grey dashed line is lower bound estimate).** Note that this refers to the volumes of carbon potentially disturbed but there remains large uncertainty concerning how much of that carbon will be remineralized, and hence lost. Data based on open access records by Telegeography (https://www.submarinecablemap.com/).

in sediments is labile, and may not be remineralized after disturbance[37]. Previous studies have attempted to calculate a mean global oxidation rate; however, there is significant variability, due in a large part to controls exerted by ocean depth, deposition rate and primary productivity, resulting in large uncertainties[38]. The degradability of organic carbon, and hence remineralization rates, strongly depend on the physiographic environment and the associated chemical, biological and physical processes[38–40]. For example, regional differences in water column and sediment oxygen concentrations, and hence markedly different carbon remineralization rates, may occur in different areas, such as coastal hypoxic zones that will feature very low remineralization rates[41]. The rate of reactivity can vary over at least four orders of magnitude in marine sediments worldwide[42]. Third, areas where cables are buried may already have been extensively trawled; hence, seabed carbon stocks may already have been perturbed. Finally, cable burial differs from bottom fishing as it is intended to be a one-off activity, in contrast to fishing that leads to repeated exhumation[3,15]. However, in the absence of field studies related to cable burial we consider the ranges of carbon loss determined from fishing studies to be a reasonable analogue for a first-order global calculation, where most carbon loss is observed following the first fishing trawl[3,15,31].

## Comparison of disturbed sediment volumes with those from other natural processes and human activities

Our results indicate that to-date up to 1.22 km³ of sediment may have been disturbed during the burial of telecommunications cables (Table 1). This is of similar magnitude to that disturbed by natural processes that can exhume equivalent quantities of sediment in individual events, such as landslides that occur on the continental slope. For example, earthquake-triggered landslides can be particularly large, such as the slope failure at the head of the Kaikōura Canyon (estimated at 1.21 km³ and 7 Mt of organic carbon) following the $M_w$ 7.8 earthquake offshore New Zealand in 2016[43] and the $M_w$ 9.0 Tohoku-oki earthquake in 2011 that triggered a 0.2 km³ landslide that transported just under 1 Mt of organic carbon to 7 km water depth[11]. In 2020, a major river flood from the Congo River triggered an underwater avalanche that transported 2.68 km³ of sediment with 3–4% organic carbon content to 5 km water depth[44]. Such natural events can also be far larger, such as the >100 km³ Grand Banks landslide, which was triggered by a $M_w$ 7.2 earthquake offshore Newfoundland in 1929[45]. These natural disturbance events are increasingly recognized as playing an important role in the fate of sedimentary organic carbon, as they may funnel carbon to become efficiently buried in deep sea fans or hadal trenches, but may also exhume previously-buried carbon that can become remineralized[11,46]. A fundamental difference, however, is that such events are part of a natural spectrum that cannot be controlled, while human activities can be modified to minimize the potential for carbon disturbance.

It is increasingly apparent that human activities may exert a greater role than natural processes with regards to disturbance and remobilization of both sediment and carbon[4,28,47]. In 2015 alone, the global production of sediment by human activities was estimated at 150 km³, which is forecast to increase in future[28]; however, this value only included dredging (5.5 km³) in marine settings and neglected other offshore activities that disturb the seabed. Subsequent studies indicate that significantly larger sediment volumes (c.50 km³ annually) may be disturbed by bottom trawling; equivalent to a sedimentary disturbance of up to 21,870 Mt/year[4,17,48]. Our upper annual sediment disturbance estimate of 0.04 km³ by cable burial is considerably smaller than the values for these marine activities; primarily due to the smaller areal footprint (i.e. trawling grounds cover $4.9 \times 10^6$ km²[2,13] compared to $3–6 \times 10^2$ km² for cable burial) despite the greater depth of penetration below seafloor of cable burial. The sedimentary organic carbon lost due to bottom trawling on an annual basis has been estimated to be >60 Mt (conservatively assuming only the top 1 cm is disturbed and 30% is lost[17]), which is at least two orders of magnitude greater than the cumulative total organic carbon lost due to cable burial since modern fiber-optic cables have been laid (Fig. 5). While the quantities of organic carbon lost due to cable burial are orders of magnitude smaller than associated with other human activities such as deep-sea trawling and dredging, they are non-trivial amounts that are not currently included in any global calculations and they add to the complex manner in which humans have and continue to alter natural sedimentary systems[47]. In light of ongoing efforts to more effectively manage marine carbon budgets, it is critical to limit disturbance of any sedimentary carbon stocks where possible. We therefore now discuss approaches that may limit such disturbance.

## Location of cable burial relative to organic carbon burial hotspots

Cable routes on the continental shelf generally do not cross many of the regions that host the highest sedimentary carbon stocks (Fig. 1B). This largely reflects the fact that most existing subsea cable routes occur in low to mid latitudes, while many of the high sedimentary carbon hotspots are focused in the Arctic, which is not presently a well-developed region for telecommunication cable routes[49]. Similarly, many other sedimentary carbon hotspots, such as offshore Namibia, Peru and Baja California, are rarely crossed. Cable routes in the Baltic Sea appear to coincide with higher than average sedimentary carbon stocks, with other notable hotspots traversed in Southeast Asia (Fig. 4). Greater constraint on the potential for the mineralization of disturbed carbon is required. In particular, the mapping of areas that are most vulnerable to carbon loss should be the focus of future studies[50].

The global carbon sediment stock calculations used here are based on a machine learning model trained using n = 11,578 sediment cores, that provides an output prediction surface with a horizontal

resolution of 5 × 5 arc minute (c.5–9 km). As a consequence of this relatively coarse spatial resolution, the global model does not include many very localized hotspots of sedimentary carbon enrichment. These carbon hotspots include coastal ecosystems, such as mangroves and seagrass meadows, and deep-sea submarine canyons[51–54]. However, organic carbon hotspots such as submarine canyons are avoided where possible for subsea cables as their irregular and steep terrain poses a hazard for routing, and they are more prone to natural hazards such as submarine landslides and sediment avalanches that can damage cables[18]. Of the total length of subsea cables, only 2.8% crosses submarine canyons (Fig. 1B) and are dominantly surface laid, rather than buried due to the water depths in which they lie; hence, any disturbance will be minor[21]. Mangroves and seagrass meadows are also avoided by cable routing where possible, primarily due to the sensitive ecology they support; however, where it is necessary to cross such areas, remedial measures can be applied, such as removal of seagrass from routes and replanting after cable emplacement, sowing of seagrass seed, or use of a custom vibrating plough in salt marshes that was shown to limit loss of sediment from the trench and where full recovery occurred within five years[55]. Directional drilling has been used to install cables beneath sensitive coastal areas to completely avoid any disturbance[56]. Blue Carbon systems such as seagrass meadows and mangroves are not incorporated in the global model used here; hence, more local assessments would assist with route planning. Near-seafloor deposits with extremely high organic carbon contents, such as buried peats in the North Sea (up to 50% total organic carbon content) may be especially vulnerable to disturbance; yet little work has been done to constrain their extent or the consequences of large losses of irrecoverable carbon from these long-term stores[57–59]. Past peaks in demand for digital connectivity saw the installation of greater numbers and lengths of subsea cables (e.g. the late 1990s "dot-com boom"; Fig. 5). Future demand for high bandwidth connectivity may therefore see a similar expansion of the submarine cable network, including new routes in areas that may feature high sedimentary carbon stocks. We suggest that the potential disturbance of sedimentary carbon stocks, and particularly minimizing the impacts on carbon hotspots, should be a consideration when planning cable routes, in a similar manner to the assessment of natural hazards, human activities and environmental impacts.

## Future opportunities to minimize disturbance of sedimentary carbon stocks

Given the minor environmental impacts of surface-laid cables[25–27], it could be tempting to suggest that cable burial is avoided to minimize disturbance of sedimentary carbon stocks. However, the primary role of cable burial is for protection against human activities, in particular bottom trawling. Indeed, the main reason cables are buried in water depths up to 2000 m is due to the expansion of bottom fishing. If bottom fishing efforts were reduced (e.g. limiting the depth of penetration of otterboards) or restricted close to cable routes, this would provide a two-fold benefit for the preservation of sedimentary carbon: (i) limiting the depth and intensity of disturbance by bottom trawling[60,61]; and in addition, (ii) reducing the depth or avoiding the need for cable burial entirely in deep water. A cable protection zone in the Cook Strait, New Zealand covers an area of 236 km² within which neither anchoring nor fishing are permitted[36]. Such initiatives elsewhere may provide mutual benefits for cable resilience and for marine ecology as they can create a reserve effect by restricting other human activities.

Unlike deep sea fishing, that may repetitively affect a fishing ground, cable burial is intended to be a one-off activity; hence, any disturbance is generally restricted to that initial period, with the exception being unpredictable and rare instances where repair is required. Historically, when they reach the end of their operational life, cables have been left in place due to their inert nature. Indeed, recovered sections of cables from the Pacific and Atlantic have

been observed to be near-pristine and physically intact after almost 50 years[62]. This limited degradation supports the case for leaving cables in place; however, these same properties make decommissioned cables potentially high-grade recycling targets, particularly the polyethylene plastic, steel and copper components. While this is potentially a valuable contribution to enhanced sustainability, assessing whether a decommissioned cable is to be recycled or left in place should carefully consider any adverse impacts its recovery may have on the seafloor environment, benthic communities[21] and sedimentary carbon stocks. While the volumes of sediment and carbon disturbance that we estimate in this study may be significantly smaller than activities such as fishing, it is important to constrain the effects of any human activity that may disturb the sedimentary carbon stocks and provide marine-planning guidance to minimize disturbance where it is possible. The impacts to carbon stores have largely been ignored for other human activities that involve excavation of seafloor sediments, such as the burial of oil and gas pipelines, cables that transfer electricity, and large diameter foundations for offshore renewables structures, but should be considered in future, particularly as infrastructure developments extend into sedimentary carbon hotspots such as the Arctic[49].

The growing demand for high-bandwidth international communications and data transfer means that the global network of subsea cables continues to grow rapidly, including new routes to previously-unconnected regions and enhancement of existing connections. These connections play a critical role in sustainable development, and reduce reliance on both domestic and international travel. While the physical environmental impact of these cables is relatively minor, we have shown that the total volume of sedimentary disturbance due to cable burial (albeit over multiple decades) can be equivalent to that exhumed during major natural disturbances, such as a submarine landslide. We highlight opportunities to minimize carbon loss for future cable routes and propose that restricting fishing activity near cable routes may have a two-fold benefit, both diminishing carbon lost due to bottom trawling and reducing or avoiding the need to bury cables in deep water. This study presents a global assessment of the sedimentary carbon that may have been disturbed by cable burial, but the uncertainties in our estimates underline a pressing need for field and laboratory-based calibration studies to determine the fate of disturbed organic carbon. Such studies are essential to constrain organic carbon disturbance and loss across a wide range of water depths, and diverse physiographic and oceanographic settings, to quantify the true loss and vulnerability of sedimentary organic carbon to human activities.

## Methods
### Types of cable burial technique

The depth and width of seafloor disturbance depends upon the requirements for protection, the nature of the seafloor substrate, and the type of cable burial technique that is used (Fig. 2). On the basis of prior studies, including field studies of pre- and post-installation disturbance, we assume a range of credible burial depths (0.5–2.0 m) and width of seafloor disturbance (0.5–1.0 m) based on published values[18,21]. We now provide details on the different types of cable burial technique.

Ploughing involves simultaneously laying and burying a cable and is a widely used technique[21]. Ploughs are towed by a cable laying ship and include an assembly mounted on skids or caterpillar tracks from which a narrow furrow is excavated by a blade ('plough share') to the desired depth of burial, which can be up to 3 m below seafloor. Excavated sediment is then allowed to fall back in and infill the furrow. The physical width of the largest commercial plough share that cuts the furrow is 45 cm, (SubCom, Pers. Comm.), but is typically 30 cm[55]. The plough skids on either side of the furrow may be up to 75 cm wide, and may compress seafloor sediments, the extent to which that occurs depends on their stiffness. The final width of the excavated furrow also

depends on sediment type. For example, a 45 cm-wide furrow might widen in poorly consolidated sediments if the sidewalls of the furrow collapse into the furrow itself. In cohesive sediments, the furrow width is likely equal to the width of the plough share, however, in granular sediments the width may be slightly wider (<1 m[21]). Ploughing is limited by the length of umbilical and tow wire to a maximum of 1200–1500 m water depth, where the plough becomes hard to control, especially coming down or up slopes over the continental shelf break.

Jetting is typically used where seabed conditions are unfavourable for ploughing, such as on steep slopes or in water depths beyond 1200–1500 m, although the latter case is rare[21]. This technique involves fluidization of the seafloor beneath a "sword" that is deployed from a Remotely Operated Vehicle that may extend to a depth of 3 m below seafloor, to fluidize a 15–30 cm-wide area. The cable is covered by sediment that settles out from the fluidized slurry. Jetting in cohesive sediments tends to create steep-sided profiles, while broader profiles are more common in granular sediments. Observed seafloor disturbance widths are <1 m, and typically much narrower[21]. Jetting disturbs the sediment far more than ploughing, and may create berms of sand and gravel close to the trench (<100 m) and can disperse suspended mud more widely (up to 2 km[21]).

Less common is the use of mechanical trenchers, where a tracked vehicle uses a mechanical chain excavator or rock cutters to excavate a trench in areas of rocky seafloor. Trenching may reach depths of up to 1.5–3 m below seafloor, with widths of <1 m. Cable routing generally avoids areas of rocky substrate, and given the expense and environmental impacts of trenching, this is a last resort. In many cases where rocky seafloors must be traversed, cables are laid in protective casing on the seafloor rather than attempting burial[21,27]. In nearshore zones where cable burial may not be possible due to challenging substrates or due to the presence of sensitive habitats (e.g. seagrass, mangroves) horizontal directional drilling may be used. This approach involves the subsurface drilling of a hole, through which a cable is passed and avoids any seafloor disturbance[21].

### Calculating the length of fiber-optic telecommunications cables in the ocean

The total length of submarine telecommunications cables was determined by summing the total length of all of the individually identified cable sections in a proprietary database provided for this project by Global Marine Ltd. This database details precise cable locations, including operational cables and those that have been decommissioned (out-of-service cables). Cross-checking this length against an open-access database of cable lengths (Telegeography: https://www.submarinecablemap.com/), indicates a difference of less than 3%, with a total length calculated from the Global Marine database of $1.82 \times 10^6$, compared to $1.88 \times 10^6$ from Telegeography. Of the total length in the Global Marine database, 13.6% of the cable length ($2.47 \times 10^5$) was reported to be out-of-service as of the December 2020. As the Telegeography database does not provide precise location information, we necessarily use the Global Marine database to calculate the length of cable that requires burial. An estimated 13.5% of the total length lies within Areas Beyond National Jurisdiction.

### Calculating the volume of disturbed sediment and carbon along cable routes

In order to calculate the volume of sediment disturbed by cable burial activities, we first determine the length of cables that are laid in water depths where burial is required. We use the 2022 GEBCO bathymetric map of the oceans (GEBCO, 2022) to determine water depths along each of the cable routes in the Global Marine database. We first excluded all cable lengths that lie in water depths >2000 m. We then differentiated by cable lengths that lie on the continental shelf, the continental shelf between to water depth of 1500 m, and between 1500 m and 2000 m (based on the World Seafloor Geomorphology map

of GRID Arendal[63]. We make this differentiation because cables are typically buried to water depths of up to 1500 m, but in some regions (particularly the NE Atlantic) burial is sometimes required to 2000 m water depth. In so doing, we aim to provide a conservative upper bound (i.e. including water depths of up to 2000 m). We then relate these cable lengths to the dimensions of the trenches excavated for cable burial, which provide upper and lower bounds for the potentially disturbed volume of sediment. Disturbed seabed area is derived by multiplying cable length by trench width (0.5–1.0 m), and then related to disturbed sediment volume by multiplying that value by trench depth (0.5–2.0 m). Finally, we relate the disturbed sediment volumes to the global modeled sedimentary carbon stocks of Atwood et al.[2]. We do this in two ways. First we simply base this on global average values of carbon/km² within the top 1 m below seafloor that Atwood et al. provide for the continental shelf and continental slope. Second, we use the mapped values of carbon/km² from the global model of Atwood (i.e. Fig. 2B), extracting the values along each cable route to enable a more geographically-resolved calculation. Where we assume a burial depth scenario of 0.5 m, we half this value, and for a burial depth of 2 m, we double the value.

### Calculating the length of cables crossing different physiographic domains

We calculate the length of cables crossing the main physiographic domains in the ocean, as well as key marine biodiversity hotspots, i.e. submarine canyons, by clipping the cable polylines in ArcGIS version 10.8 to the extent of shapefiles from the World Seafloor Geomorphology map of GRID Arendal,

### Data availability

Bathymetric data from which water depths of cable route sections were extracted are reproduced from the GEBCO_2022 Grid, GEBCO Compilation Group (2022) GEBCO 2022 Grid (https://doi.org/10.5285/e0f0bb80-ab44-2739-e053-6c86abc0289c) and can be accessed at https://www.gebco.net/data_and_products/gebco_web_services/web_map_service/. The World Seafloor Geomorphological map from which extents of continental shelves, continental slope and other physiographic domains were calculated can be accessed at https://www.arcgis.com/home/item.html?id=3a40d6b0035d4f968f2621611a77fe64. While the database of precise cable locations used for our analysis is proprietary, the approximate location of cable routes can be downloaded at https://github.com/telegeography/www.submarinecablemap.com. The global mapping of sedimentary organic carbon used in our analysis is available at https://figshare.com/articles/dataset/Global_marine_sedimentary_carbon_stock/11956356. Country outlines were sourced from Natural Earth and version 5.1.1. of the 10 m resolution open access dataset can be downloaded from https://github.com/nvkelso/natural-earth-vector/find/master.

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

## Acknowledgements
MAC acknowledges funding from the Natural Environment Research Council (NERC) National Capability Programme (NE/R015953/1) "Climate Linked Atlantic Sector Science" and from the International Cable Protection Committee. We thank Global Marine Ltd for providing access to a global database of telecommunications cables.

## Author contributions
The study was conceived by M.A.C. Analysis was led by M.A.C. A.L., S.P., and N.L.M.B. contributed to the development, writing and editing of the manuscript.

## Competing interests
M.A.C. acknowledges financial support from the International Cable Protection Committee that supported the analysis of the spatial footprint of subsea cables and the extent of their burial. All other authors declare no competing interests.
