## [Peer Review File · Nature Communications]

REVIEWERS' COMMENTS

Reviewer #1 (Remarks to the Author):

This study provides the first global estimates of length, area, sediment volume, and organic carbon disturbed by subsea cable laying. The calculations were based on the industrial database of global subsea telecommunication cable distributions and a recently published global sediment organic carbon stock map. The authors concluded that there might be 0.15-1.22 km³ of sediment and 2.04 to 16.30 Mt of organic carbon disturbed by the cable burials. Globally, 0.28 to 6.76 Mt of disturbed organic carbon may be remineralized and contribute to the ocean DIC pool. The authors estimated that the organic carbon lost due to cable laying is less than one-tenth of the carbon loss caused by bottom trawling (>60 Mt) but is still non-trivial for global marine carbon management. Finally, this study recommends limiting the depth of bottom trawling and restricting trawling in the existing and future cable areas. Such mitigation efforts would preserve the sediment organic carbon from the bottom trawling and cable laying.

Overall, this is a well-written paper, likely in the interests of the general audience. The method is straightforward but relies on reasonable estimates of sediment volume for cable burials, organic carbon concentration in the sediment, and organic carbon remineralization rate for the disturbed sediment. Unfortunately, each of these estimates is quite challenging. As a result, the high and low estimates of the sediment volume and organic carbon disturbed by cable burials, as well as the organic carbon loss by remineralization, span almost an order of magnitude difference or more. Except for the highest and lowest possible values, it is practically impossible to assess the uncertainty of these global estimates and render their applications in the marine organic budget difficult. Nevertheless, I think this paper makes a point and raises a potentially important and long-ignored source of the organic carbon disturbance and loss on the seafloor.

Reviewer #2 (Remarks to the Author):

The work presented in this paper is original, and I am very supporting to this. Particularly when it involves the deep sea which is typically characterised by scarcity of data. One may argue that the results are of low importance because they show a relatively low impact of carbon disturbance, and a (really) unknown degree of Carbon remineralisation. However it is very important to consider all possible aspects of humans activities in the marine environment however small. Particularly if the environments affected are slow in recovery, as it is the case in the deeper parts of the ocean; many of them particularly those that tend to be Carbon hotspots (either for burial or remineralisation) are classified as vulnerable. Under this light the aim is sound and the principle of the work increases our understanding that whatever we do does actually impact the marine environment; this should be taken into account when plans for any type of ocean exploitation are proposed. Therefore I find the work pioneering, certainly a small but nevertheless good step to the right direction. The final suggestion that carbon disturbance and its potential remineralisation should be taken into account for similar activities (e.g. laying of cables for wind turbines), leading eventually to some sort of establishing carbon footprints for human activities in the sea, is certainly very topical and necessary. The limitations of the work are clearly explained, namely the coarse resolution of the data and model that estimates carbon contents as well as the even bigger question about remineralisation. The approach seems sound to me although some more clarity about the calculations would help. Some suggestions are made as annotated comments in the pdf file but apart from that I have no major objections for this to work to be published. Some refinements could be still made although I doubt that will affect the result greatly. The main value of the work, as I highlighted above is that it emphasizes the need for such work and provides a very good impetus to study the deep and even coastal oceans, including transitional environments such as coastal wetlands, with a higher intensity and better resolution.

Reviewer #3 (Remarks to the Author):

The manuscript "What is the impact of the global subsea telecommunications network on sedimentary organic carbon stocks?" provides an important overview of a currently unrecognized source of anthropogenic disturbance on the seafloor and potentially impact that the installation of sub sea cables could have on the carbon stored with the sediments.

The authors estimate that between 2.82 - 11.26 Mt of organic carbon has been disturbed to to cable installation in waters to a depth of 2000m. Though this is small amount of carbon it is important to understand and account for all anthropogenic seabed disturbance.

This work is important and required yet I believe the data currently available (as discussed in the manuscript) is not suitable to produce accurate or robust estimates of carbon disturbance and possibly remineralization at the global scale. Therefore I believe this manuscript would make a better perspective article highlighting the issues, our current understanding (or lack of) and the research questions that need to be answered to get a grip on the carbon impacts of subsea cable installation.

Below I have some technical questions and comments on the data used in this study:

This study assumes a range of cable burial depths ranging between 0.5 and 2m (Line 87). The sedimentary carbon data used is from Atwood et al., 2020 is only to a depth of 1m, how do the authors extrapolate the data to a depth of 2m ? The Atwood et al., 2020 data is the only current data set that quantifies carbon in the top 1m of sediment but much of the data used in these stock estimates only represent the top 30-50cm of the sediments. The 1m carbon stocks in Atwood et al., 2020 are calculated from this data which potentiality results in a overestimation of carbon stored in the sediments, if this study further extrapolates the Atwood et al., 2020 data to 2m depth these errors will be significant.

Line 118: The authors use remineralization rate of between 20-60% taken from the literature. These values may not be appropriate for sediments globally. The Sala et al., rates are widely discredited in the literature. While the rates reported by Paradis and De Borger are much better they are geographically constrained (Mediterranean and North Sea) and there relevance at a global scale needs to be discussed further. In the preceding lines the difficulties of accurately estimating these rates are discussed I would consider if with the currently limitations the text should purely focus on the sediment/carbon disturbance and not the loss.

Annually it is estimated the sediments bury around 150 Mt of carbon it would be useful to assess potential recover times and highlight area more at risk (deep sea - slower accumulation rates). For example TAT-8 was the first fiber optic subsea cable installed in 1988 using this as a start date and 2020 as the date of the carbon map I would estimate that annually that between 0.09 and 0.35 Mt of OC per year is disturbed. This is of course crude but provides important context. This could be broken up into the different depositional areas to provide greater insights and potentially highlight the areas at greatest risk.

Ocean acidification is mentioned a few times as potential impact of disturbance of seabed organic carbon though true this could potentially happen, care must be taken to consider that the seabed holds large quantities of inorganic carbon that once disturbed would buffer against changes in pH for a significant period.

Again I think this manuscript is important for highlighting a currently unrecognized anthropogenic impact on the seabed but the lack of data at the global scale hinders accurate quantification of carbon disturbance and losses therefore I feel that the manuscript should be reworked into a perspective article.

Response to Reviewers

We thank the reviewers for their thoughtful reviews and we are pleased to receive such supportive reactions to our study, with three reviewers commenting this is “a well-written paper, likely in the interests of the general audience”, raising “a potentially important and long-ignored source of the organic carbon disturbance and loss on the seafloor” (Reviewer 1), agreeing that “the work presented in this paper is original” and “pioneering” (Reviewer 2) and “provides an important overview of a currently unrecognized source of anthropogenic disturbance on the seafloor” (Reviewer 3).

We have made a number of changes to address the reviewer comments that include:

- Adding further commentary on the fact that the global distribution of sedimentary carbon stocks is a model, based on relatively sparse, albeit >11,000 sampling points and that the model only extends to 1 m; hence (in the absence of another global model) we extrapolate the same concentration values down to 2 m (the maximum depth of cable burial assumed in this study).
- Explaining how these uncertainties may affect our results, and highlighting the need for future studies (field and laboratory) to better constrain the disturbed volumes.
- Reducing the focus on “lost carbon”, given the uncertainties and reliance on few field studies that examined the effects of bottom trawling. We have now removed the values of carbon loss from Table 1 and instead focus on the disturbed sediment and carbon volumes alone.
- Expanding upon uncertainties in remineralization rate across the global ocean as a result of varying biological, chemical and physical conditions in the Discussion, including citation of relevant modelling and field studies that provide further details on the complexity of the system.
- Adding in further clarifications concerning the limitations of the global sedimentary carbon model (e.g. it does not include local Blue Carbon systems such as mangroves or seagrasses), modified language around “potential” enhanced ocean acidification etc.

Below are the responses to each individual reviewer comments, starting with their high-level comments followed by a point by point basis to their more detailed comments. Below the reviewer comments are in *italics* and our responses in **bold**. We refer below to line numbers in the ‘clean’ version of the manuscript.

Responses to Reviewer #1

This study provides the first global estimates of length, area, sediment volume, and organic carbon disturbed by subsea cable lying. The calculations were based on the industrial database of global subsea telecommunication cable distributions and a recently published global sediment organic carbon stock map. The authors concluded that there might be 0.15-1.22 km³ of sediment and 2.04 to 16.30 Mt of organic carbon disturbed by the cable burials. Globally, 0.28 to 6.76 Mt of disturbed organic carbon may be remineralized and contribute to the ocean DIC pool. The authors estimated that the organic carbon lost due to cable laying is less than one-tenth of the carbon loss caused by bottom trawling (>60 Mt) but is still non-trivial for global marine carbon management. Finally, this study recommends limiting the depth of bottom trawling and restricting trawling in the existing and future cable areas. Such mitigation efforts would preserve the sediment organic carbon from the bottom trawling and cable laying.

Overall, this is a well-written paper, likely in the interests of the general audience. The method is straightforward but relies on reasonable estimates of sediment volume for cable burials, organic carbon concentration in the sediment, and organic carbon remineralization rate for the disturbed sediment. Unfortunately, each of these estimates is quite challenging.

As a result, the high and low estimates of the sediment volume and organic carbon disturbed by cable burials, as well as the organic carbon loss by remineralization, span almost an order of magnitude difference or more. Except for the highest and lowest possible values, it is practically impossible to assess the uncertainty of these global estimates and render their applications in the marine organic budget difficult. Nevertheless, I think this paper makes a point and raises a potentially important and long-ignored source of the organic carbon disturbance and loss on the seafloor.

We agree that this study has the potential to set a new research agenda and primes future data acquisition and research that aims to provide greater spatial coverage and depth resolution of sedimentary carbon stocks, (as also recognised by Reviewer 2, see below). We were pleased that this reviewer recognises that we identify a “potentially important and previously-ignored source of organic carbon disturbance and loss” – this was our primary aim, as well as identifying opportunities to mitigate this potential loss in future.

We thank the reviewer for their positive appraisal of our work. As they correctly state, there remains a paucity of detailed sedimentary carbon data on a global basis; however, we consider that this is not a reason not to perform the analysis (which they also support). We use the most appropriate data that are available, but now provide an expanded commentary on the limitations of the existing data, and hence also of uncertainties that may propagate into our estimates. We have modified the text to ensure that we are more careful to highlight these propagated uncertainties, including the following revisions:

- **In relation to extrapolation of the Atwood et al. dataset from 1 to 2m below seafloor: Line 103** “In the absence of any global dataset that extends below one meter, we necessarily assume a similar concentration of organic carbon exists to a depth of two meters (i.e. the maximum depth of cable burial assessed here). We accept this may result in an over-estimated disturbed carbon stock for that lower meter, and this data gap clearly underlines a need for greater constraint by future studies.”
- **In reference to speculative assessment of carbon loss: Line 133** “These remineralization rates were highest in areas affected by the greatest frequency of bottom trawls; however, as cable burial is a one-off activity, the highest remineralization rates are considered to be unlikely. Speculatively assuming the lowest loss rate (i.e. 20%) from these studies, would result in a cumulative loss of 0.144-1.17 Mt of previously-buried organic carbon on the continental shelf and 0.136-1.09 Mt on the continental slope (a total of 0.280-2.25 Mt globally). However, to date no study has specifically studied the effects of cable burial on carbon disturbance at field-scale and the whether findings from bottom trawling are truly applicable to cable burial remains unclear. Consequently, there remains considerable uncertainty in the fate of sedimentary organic carbon disturbed by cable burial” **We also now also remove the carbon loss from Table 1 as it is a more speculative estimate (see also response to other reviewers).**
- **In further reference to the uncertainties in carbon loss: Line 155** “Previous studies have attempted to calculate a mean global oxidation rate; however, there is significant variability, due in a large part to controls exerted by ocean depth, deposition rate and primary productivity, resulting in large uncertainties⁵². The degradability of organic carbon, and hence remineralization rates, strongly depend on the physiographic environment and the associated chemical, biological and physical processes⁵²⁻⁵⁴. For example, regional differences in water column and sediment

oxygen concentrations, and hence markedly different carbon remineralization rates may occur in different areas, such as coastal hypoxic zones that will feature very low remineralization rates⁵⁸. The rate of reactivity can vary over at least four orders of magnitude in marine sediments worldwide³⁴.”

- **New text to highlight potential differences in carbon loss associated with different burial types: Line 147** “A particularly important control is likely to be the cable burial tool that is used, and the nature of the initial disturbance. In the case of ploughing and trenching, sediment typically settles quickly (particularly granular sediment, such as sand) and deposits close to the initial excavation site; in many cases immediately (fully or partially) backfilling the trench²¹. In such cases, the likelihood of remineralization will be reduced; however, in the case of jetting (which fluidizes the sediment), suspended plumes of fine (clay and silt-size) sediment may be more widely dispersed by ocean currents, taking days to settle and hence increasing the chances of remineralization^{21,49}.”
- **We also add the following text to the caption for Figure 5** “Note that this refers to the volumes of carbon potentially disturbed, but there remains large uncertainty concerning how much of that carbon will be remineralized and hence lost”
- **Conclusion emphasises this uncertainty and the need for future studies to provide greater constraint: Line 296** “This study presents the first assessment of the sedimentary carbon that may have been disturbed by cable burial, but the uncertainties in our estimates underline a pressing need for field and laboratory-based calibration studies to determine the fate of disturbed organic carbon. Such studies are essential to constrain organic carbon disturbance and loss across a wide range of water depths, and diverse physiographic and oceanographic settings, to quantify the true loss and vulnerability of sedimentary organic carbon to human activities.”.

Responses to Reviewer #2:

The work presented in this paper is original, and I am very supporting to this. Particularly when it involves the deep sea which is typically characterised by scarcity of data. One may argue that the results are of low importance because they show a relatively low impact of carbon disturbance, and a (really) unknown degree of Carbon remineralisation. However it is very important to consider all possible aspects of humans activities in the marine environment however small. Particularly if the environments affected are slow in recovery, as it is the case in the deeper parts of the ocean; many of them particularly those that tend to be Carbon hotspots (either for burial or remineralisation) are classified as vulnerable. Under this light the aim is sound and the principle of the work increases our understanding that whatever we do does actually impact the marine environment; this should be taken into account when plans for any type of ocean exploitation are proposed. Therefore I find the work pioneering, certainly a small but nevertheless good step to the right direction. The final suggestion that carbon disturbance and its potential remineralisation should be taken into account for similar activities (e.g. laying of cables for wind turbines), leading eventually to some sort of establishing carbon footprints for human activities in the sea, is certainly very topical and necessary. The limitations of the work are clearly explained, namely the coarse resolution of the data and model that estimates carbon contents as well as the even bigger question about remineralisation. The approach seems sound to me although some more clarity about the calculations would help.

We were pleased that this reviewer recognised the import of the work and that our approach is sound and the limitations are clearly explained; however, as outlined in our detailed response to Reviewer 2 below, we have further strengthened the commentary on limitations. With regards to their request on some further clarity on the calculations, we have expanded the methods section, adding the following text to more clearly explain the methods:

Line 396 “Calculating the length of fiber-optic telecommunications cables in the ocean

The total length of submarine telecommunications cables was determined by summing the total length of all of the individually identified cable sections in a proprietary database provided for this project by Global Marine Ltd. This database details precise cable locations, including operational cables and those that have been decommissioned (out-of-service cables). Cross-checking this length against an open-access database of cable lengths (Telegeography: <https://www.submarinecablemap.com/>), indicates a difference of less than 3%, with a total length calculated from the Global Marine database of 1.82×10^6 , compared to 1.88×10^6 from Telegeography. Of the total length in the Global Marine database, 13.6% of the cable length (2.47×10^5) was reported to be out-of-service as of the December 2020. As the Telegeography database does not provide precise location information, we necessarily use the Global Marine database to calculate the length of cable that requires burial. An estimated 13.5% of the total length lies within Areas Beyond National Jurisdiction.

Calculating the volume of disturbed sediment and carbon along cable routes

In order to calculate the volume of sediment disturbed by cable burial activities, we first determine the length of cables that are laid in water depths where burial is required. We use the 2022 GEBCO bathymetric map of the oceans (GEBCO, 2022) to determine water depths along each of the cable routes in the Global Marine database. We first excluded all cable lengths that lie in water depths >2000 m. We then differentiated by cable lengths that lie on the continental shelf, the continental shelf between to water depth of 1500 m, and between 1500 m and 2000 m (based on the World Seafloor Geomorphology map of GRID Arendal⁵¹). We make this differentiation because cables are typically buried to water depths of up to 1500 m, but in some regions (particularly the NE Atlantic) burial is also required to 2000 m water depth. In so doing, we aim to provide a conservative upper bound (i.e. including water depths of up to 2000 m). We then relate these cable lengths to the dimensions of the trenches excavated for cable burial, which provide upper and lower bounds for the potentially disturbed volume of sediment. Disturbed seabed area is derived by multiplying cable length by trench width (0.5-1.0 m), and then related to disturbed sediment volume by multiplying that value by trench depth (0.5-2.0 m). Finally, we relate the disturbed sediment volumes to the global modeled sedimentary carbon stocks of Atwood et al.². We do this in two ways. First we simply base this on global average values of carbon/km² within the top 1 m below seafloor that Atwood et al. provide for the continental shelf and continental slope. Second, we use the mapped values of carbon/km² from the global model of Atwood (i.e. Figure 2B), extracting the values along each cable route to enable a more geographically-resolved calculation. Where we assume a burial depth scenario of 0.5 m, we half this value, and for a burial depth of 2 m, we double the value”.

Some suggestions are made as annotated comments in the pdf file but apart from that I have no major objections for this to work to be published. Some refinements could be still made although I doubt that will affect the result greatly. The main value of the work, as I highlighted above is that it emphasizes the need for such work and provides a very good impetus to study the deep and even coastal oceans, including transitional environments such as coastal wetlands, with a higher intensity and better resolution.

We took the suggestions in the annotated document into account (see detailed response to Reviewer 2). We were also glad that the reviewer recognises the

range of uncertainties in our final calculations should be a key driver for future research into sedimentary carbon across the global ocean, from coastal to deep sea settings. We hope that this study sets a new research agenda – and is implemented in policy recommendations across a range of offshore industries.

Response to detailed comments from Reviewer 2

Line 10 - Perhaps it is important to highlight the global marine sedimentary carbon storage here, i.e. similar or exceeding soil Carbon and/or 2 or 3 x atmospheric C.

There is insufficient space in the introductory abstract paragraph to add this comment. Given the fact that multiple studies present competing values, we feel that the statement in the opening sentence of the Introduction makes this point already: i.e. “Marine sediments are the largest store of organic carbon on Earth...”

Line 32 – extra space Needs correction

Space added as requested.

Line 51 – Fig 2 cited before Fig 1

Fig. 1 is actually cited at Line 41, so this is not correct. We do however now add an earlier reference to Fig 1 in Line 38 when we first introduce the global network of telecommunications cables.

Line 71 - It is a model really based on existing data which, although extensive, is nowhere near complete

This is a very good comment. We have modified the text for clarity to say “We integrate a global database that documents the extent and locations of submarine telecommunications cables, with a model of organic carbon hosted in modern ocean sediments worldwide (that is based on interpolation between >11,000 sampling points)”

Line 73 - Only a minor comment. It may be better to 'rationalise' these questions, into objectives 1a, b, 2a, b, 3 and 4.

We considered making this change, but prefer to phrase these as questions rather than numbered objectives as we feel it reads better.

Line 124 - This is true and some information on type or state of organic matter would be good, although I am not expecting this to be easily available if at all.

The reviewer is correct. This information is not readily available; hence, we have expanded the text to reference the uncertainties and the need to understand regional to local biological, chemical and physical conditions in the environment. We have added reference to three relevant studies that expand on these points.

Line 153 “Second, organic carbon mineralization rates will depend on external factors. For example, not all organic carbon stored in sediments is labile, and may not be remineralized after disturbance³³. Previous studies have attempted to calculate a mean global oxidation rate;

however, there is significant variability, due in a large part to controls exerted by ocean depth, deposition rate and primary productivity, resulting in large uncertainties⁵². The degradability of organic carbon, and hence remineralization rates, strongly depend on the physiographic environment and the associated chemical, biological and physical processes⁵²⁻⁵⁴. For example, regional differences in water column and sediment oxygen concentrations, and hence markedly different carbon remineralization rates may occur in different areas, such as coastal hypoxic zones that will feature very low remineralization rates⁵⁸. The rate of reactivity can vary over at least four orders of magnitude in marine sediments worldwide³⁴.”

Jørgensen, B.B., Wenzhöfer, F., Egger, M. and Glud, R.N., 2022. Sediment oxygen consumption: Role in the global marine carbon cycle. Earth-science reviews, 228, p.103987.

Stolpovsky, K., Dale, A.W. and Wallmann, K., 2018. A new look at the multi-G model for organic carbon degradation in surface marine sediments for coupled benthic–pelagic simulations of the global ocean. Biogeosciences, 15(11), pp.3391-3407.

Arndt, S., Jørgensen, B.B., LaRowe, D.E., Middelburg, J.J., Pancost, R.D. and Regnier, P., 2013. Quantifying the degradation of organic matter in marine sediments: A review and synthesis. Earth-science reviews, 123, pp.53-86.

In relation to this and following points of discussion: The length of time that the disturbed sediment is exposed to oxic conditions is key. In a very rudimentary manner, without having done any work, it seems that the jetting method may be more influential in Carbon remineralisation than the other two as it will induce resuspension that may take time to settle. This would be particularly important in fine grain sediments that take long(er) to settle. So sediment type is another factor that would influence these processes. Is there any information on this (e.g. grain size or simply muddy vs sandy sediments)? If this was available even in some areas then a slightly better constraint on remineralisation may be possible, e.g. using the jetting method in muddy areas may be presumed to have the highest OM decomposition (or OC remineralisation) rates

This is a very good point and we now discuss the different types of burial tools and how jetting may create suspended plumes that can be widely dispersed and take days to settle out (and hence increasing potential for carbon remineralisation in those suspended plumes) compared to other approaches that may result in shorter periods of exposure to the oxic conditions. Following jetting in fine grained sediments, turbidity can persist for several days, depending on the duration of the whole cable-laying process. For example, at the Nysted offshore wind farm (Denmark), one month was necessary to excavate 17,000 m³ of sediment for a 10.3-km long, 1.3-m wide and 1.3-m deep cable trench. However, at any given location on a cable route, disturbance will typically persist from a few hours to a few days (Taormina et al., 2018).

The following text is now included in the manuscript: Line 147 “A particularly important control is likely to be the cable burial tool that is used, and the nature of the initial disturbance. In the case of ploughing and trenching, sediment typically settles quickly (particularly granular sediment, such as sand) and deposits close to the initial excavation site; in many cases immediately (fully or partially) backfilling the trench²¹. In such cases, the likelihood of remineralization will be reduced; however, in the case of jetting (which fluidizes the sediment), suspended plumes of fine (clay and silt-size) sediment may be more widely dispersed by ocean currents, taking days to settle and hence increasing the chances of remineralization^{21,49}.”

Line 128 - it could be specified that this would be lower if upoxic/anoxic conditions are established

We agree that this is a useful clarification. We have added the following text in the revision: “For example, regional differences in water column and sediment oxygen concentrations, and hence markedly different carbon remineralisation rates may occur in different areas, such as coastal hypoxic zones that will feature very low remineralisation rates⁵⁸”.

Line 165 - for carbon

Here we are referring to sediment only so we do not make this change.

Line 195 - I doubt that seamounts are hotspots of sedimentary carbon. They are 'oases' of biodiversity and perhaps biological productivity but there is hardly any sediment on them. If anything they are more likely to be hotspots for carbon remineralisation

We agree. This was an oversight and we have removed reference to seamounts throughout.

Line 195 - 'However' may be a better word here.

We agree and made the changes as suggested.

Line 201 - Has the global model (estimates) for sedimentary marine carbon contents included coastal blue forests? In my view this is almost a separate question. Particularly for salt marshes and mangroves that are only periodically inundated.

The global model does not include these Blue Carbon systems (as stated “As a consequence of this relatively coarse spatial resolution, the global model does not include many very localized hotspots of sedimentary carbon enrichment”). These carbon hotspots include coastal ecosystems, such as mangroves and seagrass meadows, and deep-sea submarine canyons”, but we have added the following text to further clarify:

Line 238 “Blue Carbon systems such as seagrass meadows and mangroves are not incorporated in the global model used here; hence, more local assessments would assist with route planning.”

Line 207 - is there a number of 'extremely'?

We have added the comment “up to 50% total organic carbon content”, based on study of submerged peats from the North Sea (Lippman et al., 2021)

Lippmann, T.J., Van der Putten, N.N., Busschers, F.S., Hijma, M.P., van der Velden, P., de Groot, T., van Aalderen, Z., Meisel, O.H., Slomp, C.P., Niemann, H. and Jetten, M.S., 2021. Microbial activity, methane production, and carbon storage in Early Holocene North Sea peats. *Biogeosciences*, 18(19), pp.5491-5511.

Line 208 - this may be true but insights from modern (subaerial) peat bogs would help. Intuitively submerged ones are likely to have lower reactivities though as most labile OM would be gone already.

We have modified the text as following to remove explicitly reference to liability as there is still too much uncertainty in this area and it remains a frontier research topic. We also add a reference to a recent study on Canadian peatlands about irrecoverable carbon from Holocene stores as suggested. Line 240 “Near-seafloor deposits with extremely high organic carbon contents, such as buried peats in the North Sea (up to 50% total organic carbon content) may be especially vulnerable to disturbance; yet little work has been done to constrain their extent or the consequences of large losses of irrecoverable carbon from these long-term stores^{47,48,55}.”

Harris, L.I., Richardson, K., Bona, K.A., Davidson, S.J., Finkelstein, S.A., Garneau, M., McLaughlin, J., Nwaishi, F., Olefeldt, D., Packalen, M. and Roulet, N.T., 2022. The essential carbon service provided by northern peatlands. *Frontiers in Ecology and the Environment*, 20(4), pp.222-230.

Line 224 - *On the other hand most deep water settings have very low OC contents well less than 1% of dry sediments.*

We agree this is generally the case, but in some locations there can be higher organic carbon contents. Regardless, our comment here is in reference to water depths of up to 2000 m, rather than just in deep water (so this would include shallower water higher organic carbon sediments. We have reworded the sentence for clarity: Line 255 “Indeed, the main reason cables are buried in water depths up to 2000 m is due to the expansion of bottom fishing..”

Line 257 – *major natural disturbance such as a submarine landslides*
Changed to “such as a submarine landslide”

Line 268 – *organic carbon (add space)*

We cannot see where a space needs adding. It is possible this is an artefact created when the .pdf was created, but it is not visible in our native Word copy.

Reviewer #3 (Remarks to the Author):

The manuscript "What is the impact of the global subsea telecommunications network on sedimentary organic carbon stocks?" provides an important overview of a currently unrecognized source of anthropogenic disturbance on the seafloor and potentially impact that the installation of sub sea cables could have on the carbon stored with the sediments. The authors estimate that between 2.82 - 11.26 Mt of organic carbon has been disturbed to to cable installation in waters to a depth of 2000m. Though this is small amount of carbon it is important to understand and account for all anthropogenic seabed disturbance. This work is important and required yet I believe the data currently available (as discussed in the manuscript) is not suitable to produce accurate or robust estimates of carbon disturbance and possibly remineralization at the global scale. Therefore I believe this manuscript would make a better perspective article highlighting the issues, our current understanding (or lack of) and the research questions that need to be answered to get a grip on the carbon impacts of subsea cable installation.

The reviewer clearly grasps the aims of the study and our key findings. In response to their suggestion that this paper should be a perspective article, we note that the Editor commented “We disagree with Referee #3 that this should not be a research article but a Perspective, given the original data and analyses you are implying here.”

This study assumes a range of cable burial depths ranging between 0.5 and 2m (Line 87). The sedimentary carbon data used is from Atwood et al., 2020 is only to a depth of 1m, how do the authors extrapolate the data to a depth of 2m ? The Atwood et al., 2020 data is the only current data set that quantifies carbon in the top 1m of sediment but much of the data used in these stock estimates only represent the top 30-50cm of the sediments. The 1m carbon stocks in Atwood et al., 2020 are calculated from this data which potentially results in an overestimation of carbon stored in the sediments, if this study further extrapolates the Atwood et al., 2020 data to 2m depth these errors will be significant.

This is a valid comment and we have now added text to ensure this is clear. We use the global model of Atwood et al. (2020), which is the best global model available to date; however, as remarked by the reviewer, this only extends to 1 m below seafloor. In order to infer the amount of disturbed carbon to the maximum depth of cable burial, we double the amount of carbon. We recognise that this is potentially an overestimate, but in the absence of a more appropriate model this is a best endeavours approach. We now more explicitly reference this assumption in the text and explain that this point underlines a need for further constraint and future studies.

Line 103: “In the absence of any global dataset that extends below one meter, we necessarily assume a similar concentration of organic carbon exists to a depth of two meters (i.e. the maximum depth of cable burial assessed here). We accept this may result in an over-estimated disturbed carbon stock for that lower meter, and this data gap clearly underlines a need for greater constraint by future studies.”.

Line 118: The authors use remineralization rate of between 20-60% taken from the literature. These values may not be appropriate for sediments globally. The Sala et al., rates are widely discredited in the literature. While the rates reported by Paradis and De Borger are much better they are geographically constrained (Mediterranean and North Sea) and their relevance at a global scale needs to be discussed further. In the preceding lines the difficulties of accurately estimating these rates are discussed I would consider if with the currently limitations the text should purely focus on the sediment/carbon disturbance and not the loss.

We have taken this feedback onboard and now discount the rates by Sala. Consequently, we have revised the text to make it clearer that these estimates come from the Mediterranean and North Sea and that loss rates were highest where bottom trawling was most frequent. As cable burial is a one-off activity, it is likely that the upper estimate will be excessive and so opt for the lower value of 20% to provide what we now refer to as a speculative loss. We also remove the values of carbon loss from Table 1 and reserve the discussion of these speculative values of loss to the Discussion text. We have revised the text as follows:

Line 133 “These remineralization rates were highest in areas affected by the greatest frequency of bottom trawls; however, as cable burial is a one-off activity, the highest remineralization rates are considered to be unlikely. Speculatively assuming the lowest loss rate (i.e. 20%) from these studies, would result in a cumulative loss of 0.144-1.17 Mt of previously-buried organic carbon on the continental shelf and 0.136-1.09 Mt on the continental slope (a total of 0.280-2.25 Mt globally). However, to date no study has specifically studied the effects of cable burial on carbon disturbance at field-scale and the whether findings from bottom trawling are truly applicable to cable burial remains unclear. Consequently, there remains considerable uncertainty in the fate of sedimentary organic carbon disturbed by cable burial”

We have also made modifications to the language throughout the manuscript to ensure that this uncertainty is clear, including adding the following text to the caption for Figure 5 “Note that this refers to the volumes of carbon potentially disturbed, but there remains large uncertainty concerning how much of that carbon will be remineralized and hence lost”

Annually it is estimated the sediments bury around 150 Mt of carbon it would be useful to assess potential recover times and highlight area more at risk (deep sea - slower accumulation rates). For example TAT-8 was the first fiber optic subsea cable installed in 1988 using this as a start date and 2020 as the date of the carbon map I would estimate that annually that between 0.09 and 0.35 Mt of OC per year is disturbed. This is of course crude but provides important context. This could be broken up into the different depositional areas to provide greater insights and potentially highlight the areas at greatest risk.

The reviewer is very close in their estimate. The annual average (mean) is between 0.057-0.455 Mt of organic carbon disturbed per year. We have now annotated these values to Figure 5. Because cables can cross multiple regions we have, however, opted to not to follow their further suggestion to break this down further into different depositional areas vs time as we felt this was an unnecessary complication that does not sufficiently add to the paper.

Ocean acidification is mentioned a few times as potential impact of disturbance of seabed organic carbon though true this could potentially happen, care must be taken to consider that the seabed holds large quantities of inorganic carbon that once disturbed would buffer against changes in pH for a significant period.

We have modified text accordingly, removing the comment from the abstract paragraph, and caveating with “potentially” in the Introduction so this is more circumspect.

Again, I think this manuscript is important for highlighting a currently unrecognized anthropogenic impact on the seabed but the lack of data at the global scale hinders accurate quantification of carbon disturbance and losses therefore I feel that the manuscript should be reworked into a perspective article.

See earlier response that this is appropriate for publication as a research article. As already highlighted, we have added further comment on the need for further studies that provide enhanced constraint at both laboratory and field scale to ensure this is clear.